# Observation of geometry-dependent skin effect in non-Hermitian phononic crystals with exceptional points

Qiuyan Zhou[1,4], Jien Wu[2,4], Zhenhang Pu [1], Jiuyang Lu [1,2], Xueqin Huang [2], Weiyin Deng [1,2] ✉, Manzhu Ke [1] ✉ & Zhengyou Liu [1,3] ✉

Exceptional points and skin effect, as the two distinct hallmark features unique to the non-Hermitian physics, have each attracted enormous interests. Recent theoretical works reveal that the topologically nontrivial exceptional points can guarantee the non-Hermitian skin effect, which is geometry-dependent, relating these two unique phenomena. However, such novel relation remains to be confirmed by experiments. Here, we realize a non-Hermitian phononic crystal with exceptional points, which exhibits the geometry-dependent skin effect. The exceptional points connected by the bulk Fermi arcs, and the skin effects with the geometry dependence, are evidenced in simulations and experiments. Our work, building an experimental bridge between the exceptional points and skin effect and uncovering the unconventional geometry-dependent skin effect, expands a horizon in non-Hermitian physics.

Non-Hermitian physics, featured with a complex spectrum, has become a thriving realm, as evidenced by the various implementations in condensed matter and artificial periodic structures[1–3]. Owing to the exceptional topology, the non-Hermitian systems exhibit a variety of intriguing properties, including the exceptional points (EPs) and skin effect, and give rise to novel phenomena with potential applications, such as the topological lasing[4,5], topological light steering[6], and funneling[7], and topological sound whispering gallery[8]. EPs are the spectral degeneracies of the non-Hermitian Hamiltonian, in which both the eigenvalues and eigenvectors are coalesced[9,10]. As the singular point, the EP carries a nonzero topological charge. For example, the second-order EPs with two coalesced states can possess the topological charges ±1 described by the discriminant number[10,11], and give rise to the bulk Fermi arc, which connects a pair of EPs with opposite charges[12]. The EPs have been extended from point degeneracy to ring[13,14] and surface[15] degeneracies, and generalized from second-order to third-order with exceptional arcs[16,17]. The EP is further found to be utilized to make knots with topological non-Abelian braiding[18–20].

Besides the EPs, the recent advance of non-Hermitian physics is the skin effect, which is attributed to the non-Hermitian band topology[21–28]. Skin effect is that the bulk modes collapse to the open boundaries as skin modes, whose quantity scales with the volume of the system[27–31]. This poses a challenge to the Bloch band theory and the related Hermitian band topology with bulk-boundary correspondence, thus opening a new avenue and attracting extensive research interest[21–37]. In one dimension, the skin effect exhibits that all the bulk modes localize at the boundaries, and has been confirmed in the quantum walk[38], electric circuit[39], and phononic crystal (PC)[40,41]. In two dimensions, the higher-order skin effect that all the bulk modes collapse to the corners has been proposed[42,43] and realized in the non-Hermitian electric circuit[44] and PC[45]. The skin effect was also proposed in the PC for elastic wave[46–48], which may have potential applications. It is interesting that the skin effect can be anomalous to make the topological boundary states delocalize to the extended states[49–51]. Very recently, it was shown that the skin effect can be guaranteed by the EPs with nonzero topological charge, and displays a new configuration, i.e., geometry-dependent skin effect (GDSE), which may disappear for the

[1]Key Laboratory of Artificial Micro- and Nanostructures of Ministry of Education and School of Physics and Technology, Wuhan University, Wuhan 430072, China. [2]School of Physics and Optoelectronics, South China University of Technology, Guangzhou, Guangdong 510640, China. [3]Institute for Advanced Studies, Wuhan University, Wuhan 430072, China. [4]These authors contributed equally: Qiuyan Zhou, Jien Wu. ✉e-mail: dengwy@whu.edu.cn; mzke@whu.edu.cn; zyliu@whu.edu.cn

system with a particular shape[52–54]. However, such profound connection between the EP and unconventional skin effect is yet to be confirmed in experiments.

In this work, we realize a reciprocal two-dimensional (2D) non-Hermitian PC with loss, and observe the EPs and GDSE, verifying that the system with EP has the GDSE. The PC has been proved to be a versatile platform to explore frontier physics, owning to the macroscopic scale[55]. The designed loss in the PC is induced by the holes on waveguide sealed with the sound-absorbing sponges. We first illustrate the relation between the EP and GDSE by a tight-binding model on a square lattice, and then present the experimental observations of the EPs with the 2D bulk Fermi arcs, and the GDSE in the 1D ribbon under diamond-stripe geometry and finite-size sample under diamond-shaped geometry in PCs. Skin effect disappears under the square-stripe geometry and square-shaped geometry. These theoretical, simulated, and experimental results consistently evidence the EPs and the ensuing GDSE.

## Results

### Tight-binding model

To be concrete, we first construct a tight-binding model on a square lattice in Fig. 1a, where the unit cell (light blue area) contains two different sites (A and B). The Hamiltonian in momentum space can be written as

$$H = (d_x + i\gamma \cos k_x)\sigma_x + (d_y + i\gamma \sin k_x)\sigma_y \tag{1}$$

where $d_x = (t_1 + t_2)\cos k_x + 2t_y \cos k_y$, $d_y = (t_1 - t_2)\sin k_x$, $\sigma_{x,y}$ are the Pauli matrixes. The non-Hermitian interaction is generated by the designed loss in the hopping with strength $\gamma$, while the Hermitian hoppings are $t_1$, $t_2$, and $t_y$. $\mathbf{k} = (k_x, k_y)$ is the wavevector. The distance

$a/\sqrt{2}$ between nearest-neighbor sites is set to unity, where $a$ is the lattice constant. In Hermitian case ($\gamma = 0$), the system has two Dirac points[56], but in non-Hermitian case ($\gamma \neq 0$), a Dirac point splits into a pair of second-order EPs hosting opposite charges. The EPs are singular points at which two or more eigenvalues, and their corresponding eigenvectors, coalesce and become degenerate. The second-order EP means that the degeneracy is two-fold. The order of EPs has no direct connection with the skin effect. As shown in Fig. 1b, there are two pairs of EPs in the first Brillouin zone (BZ), and each pair with opposite charges $\pm 1$ are connected by a bulk Fermi arc. The topological charge of EP can be described by the discriminant number calculated by $\nu(\mathbf{k}_{EP}) = \frac{1}{2\pi i} \oint_{\Gamma_{EP}} d\mathbf{k} \cdot \nabla_{\mathbf{k}} \ln \det[H(\mathbf{k}) - E(\mathbf{k}_{EP})]$, where $\Gamma_{EP}$ represents a closed loop enclosing an EP in momentum space anticlockwise and $E(\mathbf{k}_{EP})$ is the eigenvalue of the EP[11]. The calculated results indicate that EPs have topological charge $\pm 1$, which are denoted by red and blue spheres in Fig. 1b. The detailed properties of EPs are discussed in Supplementary S-I A-C with Figs. S1 and S2.

The relation between EP and GDSE is that the EP with nonzero topological charge can guarantee the emergence of the GDSE, but it is not vice versa. This relation is discussed in detail in Supplementary S-I D-G with Figs. S3−S7. The EPs host nonzero topological charges, but they are the sufficient and unnecessary condition for the topological charges. Then the topological charges ±1 of EPs lead to the nonzero spectral winding numbers along some straight lines (directions) in the first BZ, which is defined in Supplementary S-I F. The nonzero winding number is equivalent to nonzero spectral area covered by energy $E_m(\mathbf{k})$ on the complex plane, where $m$ denotes the band index and $\mathbf{k}$ is the wave vector. Whether the skin effect emerges at the open boundaries can be judged by analyzing the mirror symmetries of bulk Hamiltonian and calculating the winding numbers of straight lines. The ribbons under open boundaries along the direction with nonzero

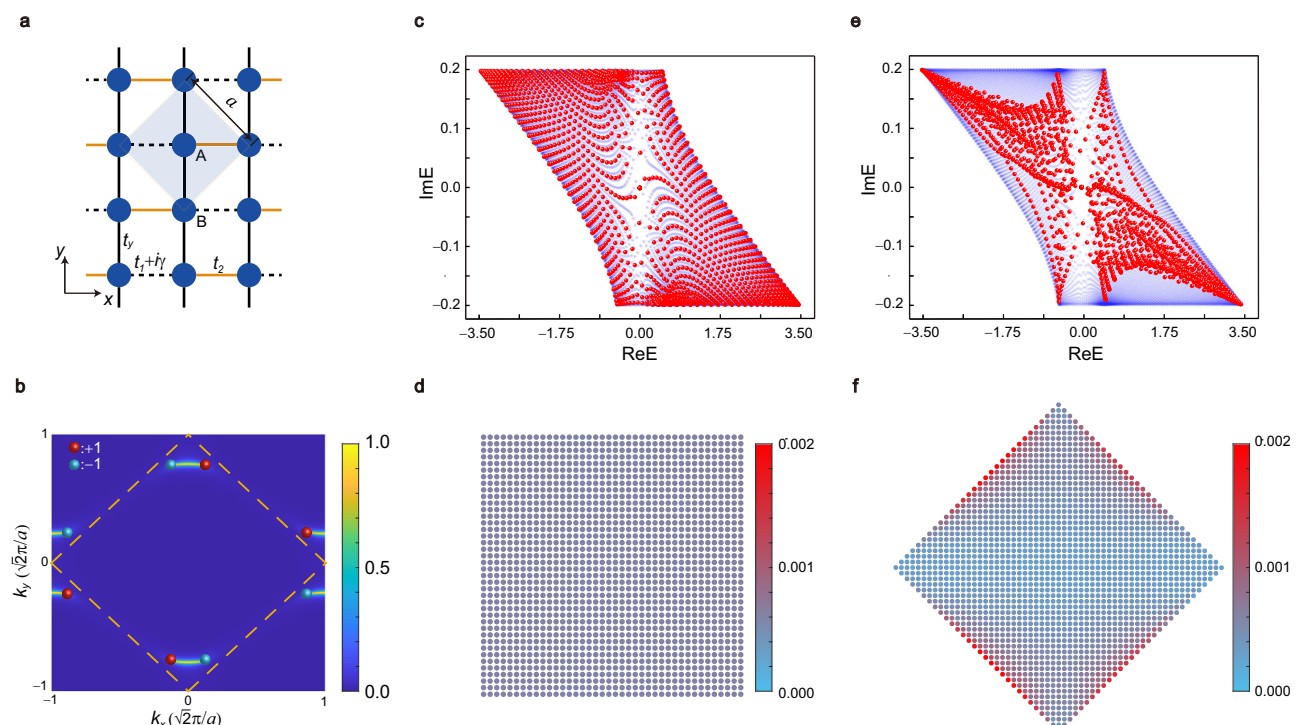

**Fig. 1 | GDSE in a non-Hermitian square lattice model with EPs. a** Schematics of the lattice structure with two inequivalent sites A and B in a unit cell (light blue area). Loss is added in the hopping $t_1 + i\gamma$ along the $x$ direction (dashed black line). $t_2$ is the Hermitian hopping along the $x$ direction (solid yellow line), and $t_y$ is the hopping along the $y$ direction (solid black line). **b** Isofrequency curve at zero energy. The yellow dashed lines denote the first BZ. Two bulk Fermi arcs locate in the first BZ and each one connects a pair of EPs with opposite charges (red and blue spheres). **c** Spectral area under square-shaped geometry with fully open boundaries (red dots). The blue background represents the spectral area under periodic boundaries. **d** Spatial distribution of eigenstates $W(j)$ under square-shaped geometry. **e**, **f** The same to **c**, **d** but under diamond-shaped geometry. The skin effect appears in **f** but disappears in **d**, thus is the GDSE. The parameters are chosen as $t_1 = t_y = -1$, $t_2 = -0.5$ and $\gamma = 0.2$.

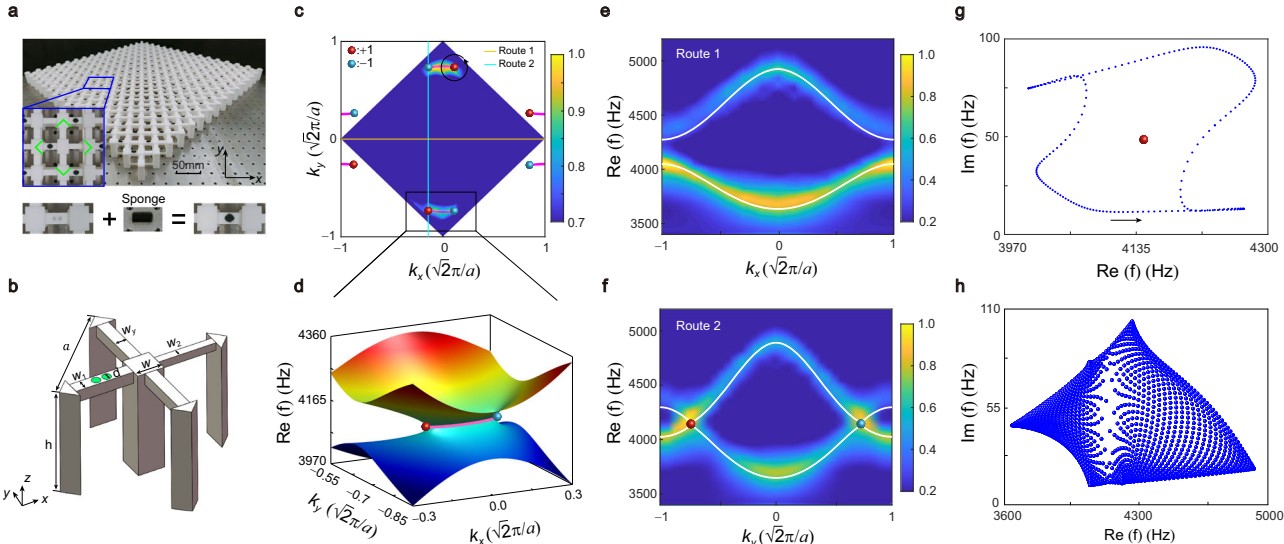

**Fig. 2 | EPs with bulk Fermi arcs and nonzero spectral area in the 2D non-Hermitian PC. a** Top panel: photograph of the sample with an enlarged part shown in the inset, where the green dashed lines enclose a unit cell. Bottom panel: Loss is induced by the holes on the waveguide sealed with the sound-absorbing sponges. **b** Schematic of the PC unit cell. **c** Simulated and measured bulk Fermi arcs at 4150 Hz. The magenta solid lines are the simulated result, which agrees well with measured one shown by color maps in the first BZ. Red (blue) spheres denote EPs with topological charge +1 (−1). **d** Calculated real part of bulk band dispersion for the range enclosed by the black framework in **c**, which exhibits a pair of EPs connected by a bulk Fermi arc. **e, f** Simulated (white lines) and measured (color map) dispersion curves for routes 1 and 2 depicted in **c**, respectively. **g** Spectral loop on the complex plane plotted along the anticlockwise direction of the black circle route around the EP in **c**. The red sphere denotes the energy projection point of the enclosed EP. **h** Complex spectra of the PC under periodic boundaries showing nonzero spectral area, giving rise to the GDSE.

spectral winding numbers host the skin effect, while those with zero winding number do not, giving rise to the GDSE in the 1D ribbons. In our system, the skin effect emerges in the ribbon under the same open boundaries to the diamond-shaped geometry, i.e., diamond-stripe geometry, but disappears under square-stripe geometry. Finally, the GDSE in the 1D ribbons can directly result in the GDSE in the finite-size samples with fully open boundaries.

Here we consider two finite-size samples under open boundaries, which are the square- and diamond-shaped geometries. The fundamental difference between these two geometries is that their open boundaries are in different directions, and more importantly, the spectral winding numbers along these directions are different, leading to GDSE. In general, the skin effect exists in the systems under open boundaries only when they have different spectral area or same spectral area but with different density distribution from that under periodic boundaries[52]. The nonzero spectral area is equivalent to the spectral winding numbers along some straight lines in the first Brillouin zone, thus always gives GDSE. The nonzero spectral area originates from the nonzero topological charge of generic point $\mathbf{k}_r$ in the first BZ, which can be degenerate or not. The topological charge is calculated as $\nu_m(\mathbf{k}_r) = \frac{1}{2\pi i} \oint_{\Gamma_r} d\mathbf{k} \cdot \nabla_{\mathbf{k}} \ln \det[H(\mathbf{k}) - E_m(\mathbf{k}_r)]$, where $\Gamma_r$ denotes a closed loop enclosing the generic point in momentum space anticlockwise and $E_m(\mathbf{k}_r)$ is the energy of the $m$-th band at the generic point. It should be noted that EP with nonzero topological charge is the typical generic point, but not the all, thus is not the only source for the nonzero spectral area. As shown in Fig. 1c, the spectral area and its density distribution (red dots) under square-shaped geometry are same as those under periodic boundaries (blue dots), and do not give rise to the skin effect. On the contrary, the spectral area under diamond-shaped geometry is different from that under periodic boundaries, as shown in Fig. 1e, leading to the skin effect. To visualize the GDSE, we calculate the spatial distribution of all eigenstates defined as $W(j) = \frac{1}{N} \sum_n |\psi_n(j)|^2$, where $\psi_n(j)$ is the $n$-th normalized right eigenstate at site $j$ and $N$ denotes the total number of the eigenstates. The spatial distributions $W(j)$ under the square- and diamond-shaped geometries are shown in Fig. 1d, f, respectively. One can see that the skin effect only

emerges under the diamond-shaped geometry, thus is the GDSE satisfying the volume law, i.e., the number of skin modes increases in proportion to the increase in the volume of the system. The geometries and volume law are discussed in detail in Supplementary S-I H and I with Fig. S8. As a result, our system having the EPs exhibits the GDSE.

## EPs and bulk Fermi arcs in the PC

We now realize the tight-binding model in a PC for acoustic waves. Figure 2a displays the experimental PC sample fabricated by 3D printing technology, where the acoustic cavity and coupling waveguide can be viewed as the site and hopping terms in the tight-binding model. The designed loss is induced by the holes on waveguide sealed with the sound-absorbing sponges, which can make a larger loss and reduce the frequency shift. The positive (negative) imaginary part of frequency indicates the attenuation (amplification) of acoustic field, and it is impossible to achieve amplification without gain in the PCs. When considering the designed loss, the intrinsic resonant frequency of each cavity will be influenced and appear global positive imaginary part which indicates attenuation. In the corresponding tight-binding model, such global loss can be described by adding positive imaginary part on the site energy, which will integrally lift the imaginary part of energy and make it positive. Since shifting the imaginary part of energy does not affect the topology property of the band, we do not consider this term in the tight-binding Hamiltonian for simplicity. The schematic in Fig. 2b is the unit cell corresponding to the enlarged view inserted in Fig. 2a. The lattice constant is $a = 45.25$ mm. The height and width of cavities are chosen as $h = 40$ mm and $w = 10$ mm, respectively. The widths of waveguides along the $x$ direction are $w_1 = 4.5$ mm and $w_2 = 3$ mm, while that along the $y$ direction is $w_y = 4.5$ mm. The diameter of holes on waveguide is $d = 3.2$ mm.

The bulk Fermi arc is terminated by a pair of EPs with opposite charges, hence can reveal the existence of EPs. The measured (color maps) and simulated (magenta solid lines) isofrequency curves at 4150 Hz are shown in Fig. 2c, visualizing the bulk Fermi arc terminated at the EPs. In simulations, we add an imaginary part on the velocity of waveguide as the designed loss (Supplementary S-II with Fig. S9).

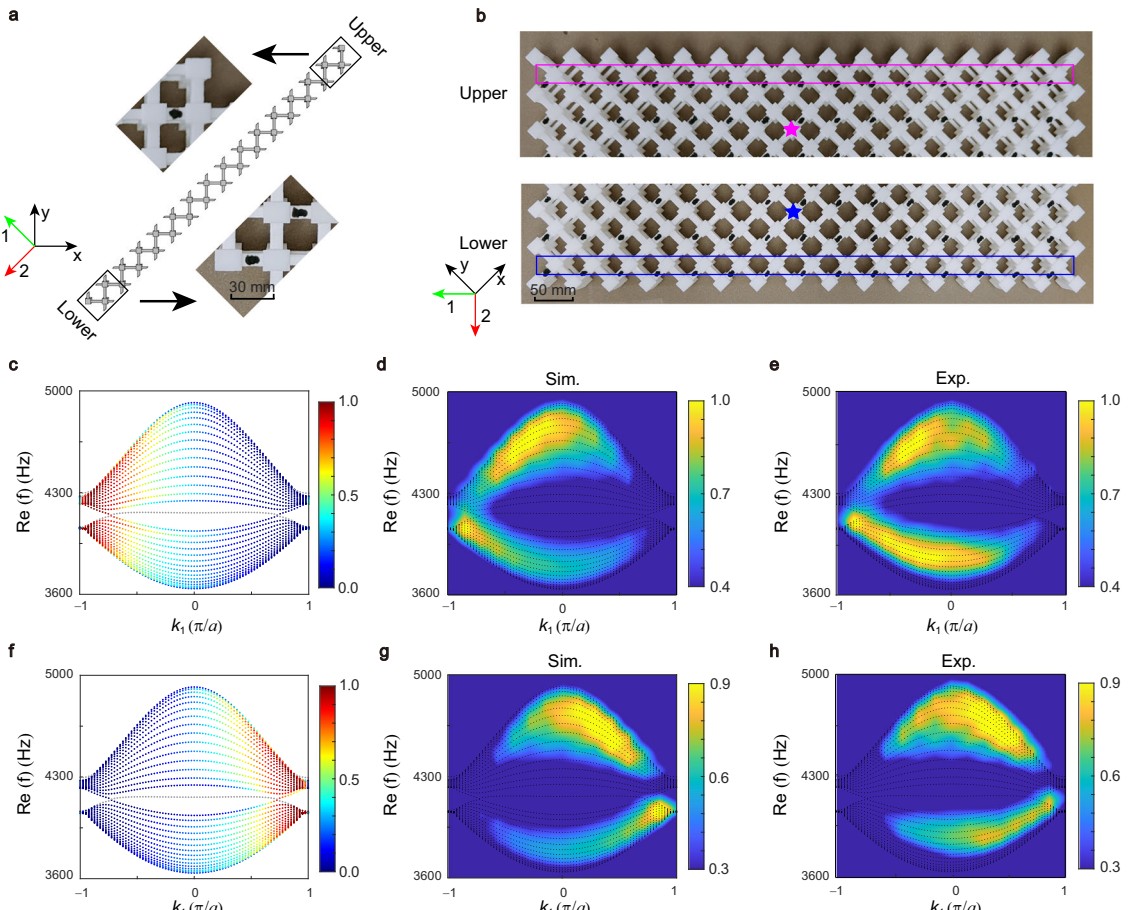

**Fig. 3 | Observation of GDSE in the 1D PC ribbon under diamond-stripe geometry. a** A sketch of the 1D PC ribbon. Insets show the experimental realization for upper and lower boundaries of the PC ribbon. **b** Photographs of upper and lower boundaries for the experimental measurement of the skin modes. Magenta (blue) star represents the position of source at upper (lower) boundary, and the response fields are measured in the magenta (blue) rectangle. **c** Real part of the projected band dispersion for the ribbon, where the color represents the localization degree of the bulk states at the lower boundary, and the gray dots denote the edge states. **d, e** Simulated and measured projected band dispersions obtained by the pressure field at the lower boundary. **f**–**h** The same to **c**–**e** but at the upper boundary. Lots of bulk states for $k_1 < 0$ ($k_1 > 0$) are the skin modes localized at the lower (upper) boundary.

The real part of bulk band dispersion near the EPs is calculated in Fig. 2d, which exhibits a pair of EPs connected by a bulk Fermi arc. We further measure the real parts of bulk band dispersion along routes 1 and 2 in the BZ (Fig. 2c), as shown in Fig. 2e, f, respectively, which together with the observation of the bulk Fermi arcs verifies the existence of EPs. More measured and simulated results, including the real part of bulk band dispersion along the high-symmetry lines and isofrequency curves at different frequencies, are shown in Supplementary S-III to S-V with Figs. S10–S12. These results consistently confirm the stable EPs connected with the bulk Fermi arc in the PC. The topological charge of EP in the PC is same to that in the lattice model, as denoted by red and blue spheres in Fig. 2c. The nonzero topological charge describes the spectral winding around the EP on the complex plane. As calculated in Fig. 2g, when choosing arbitrarily an anticlockwise route surrounding the EP with topological charge +1 (the black circle in Fig. 2c), a spectral loop winds the EP in an anticlockwise direction (arrow) on the complex plane. All the spectral loops for the EPs and other generic points finally form the nonzero spectral area, as shown in Fig. 2h, and result in the GDSE.

**GDSE in the 1D PC ribbon**

We then present the observation of GDSE in the 1D PC ribbon. Theoretical studies have shown that nonzero spectral area leads to the GDSE in the 1D ribbon under stripe geometry, and further emerging in the finite-size sample under the same open boundaries to the 1D ribbon.

The sketch of the 1D PC ribbon under diamond-stripe geometry is shown in Fig. 3a, in which its upper and lower open boundaries (Fig. 3b) are same to those in the finite-size sample under diamond-shaped geometry, and the wavevector is along the $k_1$ direction. The projected band dispersion with bulk states localized at the lower boundary is plotted in Fig. 3c. The color denotes the localization degree of eigenstates at the lower boundary, which is calculated by $D = \sum_{x \in L} |\psi(x)|^2$ with $L$ as the defined lower boundary length. One can see that a lot of the eigenstates with $k_1 < 0$ are localized at the lower boundary as the skin modes, and the degree of localization enhances as $k_1$ decreases. In experiment, we place a source (blue star) near the lower boundary to excite the skin modes there, as shown in Fig. 3b. We measure the response field of the second row of cavities near the lower boundary (blue rectangle). The measured band dispersion of skin modes localizing at the lower boundary can be obtained by the Fourier transforming the measured field. The localization degree of the skin modes is revealed by the intensity of the Fourier transformation denoted by color. The simulated and experimental results are shown in Fig. 3d, e, respectively, where the black dots represent the calculated projected band dispersion. The excited modes (the lighten area) mainly at $k_1 < 0$ are the bulk states, consistently evidencing these bulk states are the skin modes localized at the lower boundary.

Figure 3f shows the projected band dispersion with bulk states localized at the upper boundary. The color represents the localization degree of eigenstates at the upper boundary, indicating the bulk states

for $k_1 > 0$ are mainly the skin modes localized at the upper boundary. This result is also confirmed by the simulated and measured data, as presented in Fig. 3g, h, respectively. In this case, we place a source (magenta star) near the upper boundary to excite the skin modes there, as shown in Fig. 3b. To avoid the edge states located at the sublattice that is the same to the outermost cavity, we measure the response field of the second row of cavities near the upper boundary (magenta rectangle). The excited modes are the bulk states mainly focused on the $k_1 > 0$ (the lighten area), thus are the skin modes localized at the upper boundary. Although edge states arise at the open boundaries, as shown by the gray dots in Fig. 3c, f, they do not have influence on the skin effect. Hence, skin effect emerges in the PC ribbon under diamond-stripe geometry, in which the skin modes localize at the lower (upper) boundary for $k_1 < 0$ ($k_1 > 0$), but disappears in the ribbon under square-stripe geometry (Supplementary S-VI with Fig. S13). The skin effect here exhibits the geometry-dependent property, thus is the GDSE.

## GDSE in the finite-size PC sample

The existence of the GDSE can be more directly and clearly revealed by the pressure field distribution in the finite-size PC. The schematics of the PC sample with 16 cavities at the outermost boundary is shown in Fig. 4a. The spectral area of the PC sample under diamond-shaped geometry with fully open boundaries is plotted by the red dots in Fig. 4b, where the shadow area is the range covered by the spectrum under periodic boundaries. Obviously, the spectral areas under these two boundary conditions are quite different, indicating the existence of skin effect in the PC sample under diamond-shaped geometry. The eigenstates denoted by gray dots are the edge states induced by open boundaries, which do not affect the existence of the skin effect. The corresponding real part of eigenfrequency spectrum is shown in

Fig. 4c. The color represents the degree of localization for each eigenstate on open boundaries, in which the boundary area is defined as the outermost four layers cavities. The skin modes denoted by colors with larger numerical value are most concentrated in the center of frequency spectrum, while the conventional bulk modes focus on the lower and higher frequencies, consistent with those in the 1D PC ribbon. This can be verified by the response spectra in experiment. As shown in Fig. 4d, the red line ($T_{12}$) is the response spectrum of skin mode, where the source (detector) is located at point 1 (2) marked in Fig. 4a. Its peak is near the center of frequency spectrum, while those of conventional bulk modes at the lower and higher frequencies shown in the black line ($T_{34}$). $T_{34}$ is the response spectrum of bulk modes, where the source (detector) is located at point 3 (4) marked in Fig. 4a. To visualize the skin modes, we further measure the pressure field distributions for fixed frequencies. The source excites at each cavity and the response of acoustic pressure is measured at the same cavity (Methods). Figure 4e, f shows the measured pressure field distributions at 3688 Hz and 4296 Hz, respectively. One can see that the fields at 4296 Hz have stronger distributions at the boundaries, confirming the skin modes in the finite-size PC sample under diamond-shaped geometry. Since no skin mode is observed in the PC sample under square-shaped geometry (Supplementary S-VII with Figs. S14, S15), we confirm that this skin effect is GDSE.

## Discussion

In conclusion, motivated by the pioneering theoretical prediction[52], we have realized a non-Hermitian PC, which hosts two pairs of second-order EPs and exhibits the GDSE. Our work builds an experimental bridge between the EPs and skin effect, the two distinct phenomena only existing in the non-Hermitian systems, thus is of fundamental significance and paves a way for applications of non-Hermitian

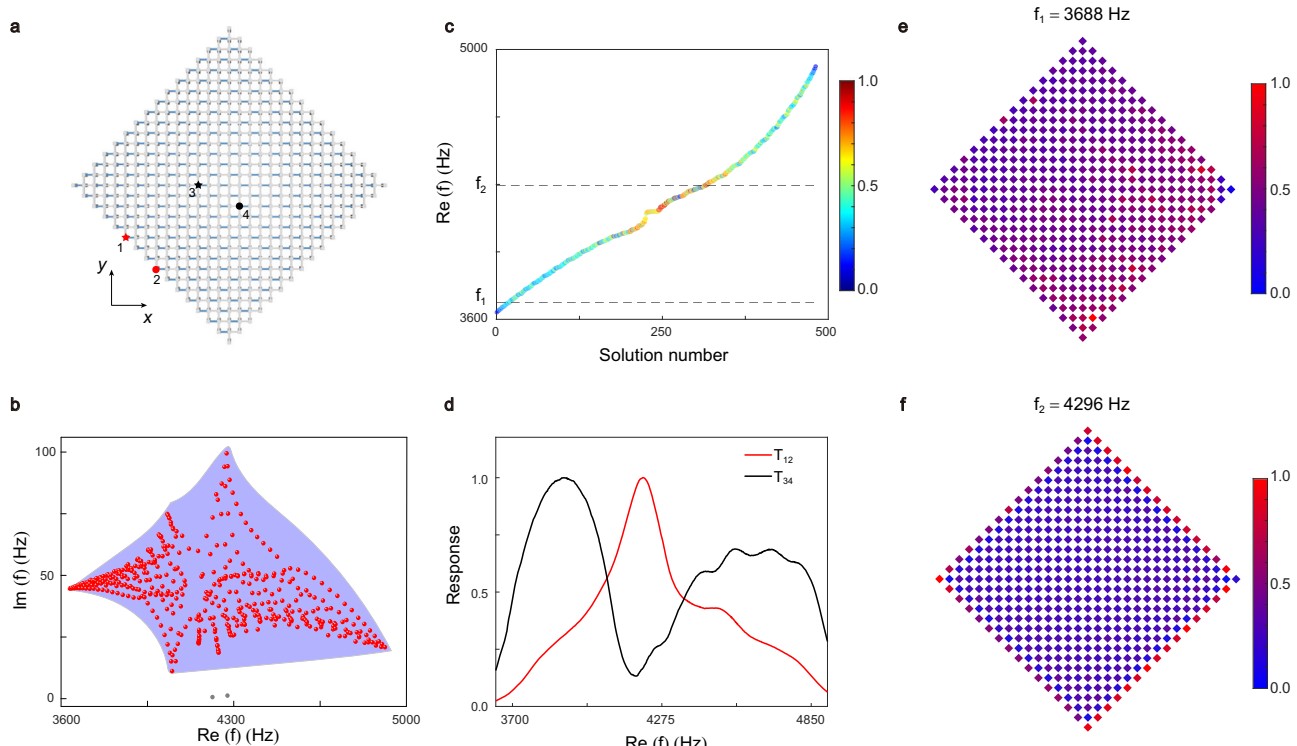

**Fig. 4 | Observation of GDSE in the finite-size PC sample under diamond-shaped geometry. a** Schematics of the finite-size PC sample. **b** Spectral area (red dots) for the diamond-shaped PC. Blue shadow area represents the spectrum under periodic boundaries, and gray dots denote the edge states induced by open boundaries. **c** Real part of eigenfrequency spectrum. The color represents the degree of localization for each eigenstate on open boundaries. **d** Measured response spectra for the skin modes ($T_{12}$) and conventional bulk modes ($T_{34}$), which are normalized by their maximum values. **e, f** Measured pressure field distributions at 3688 Hz and 4296 Hz, respectively. The fields in **f** mainly focus on the boundaries, visualizing the skin modes.

topological acoustics. With the flexibility in realizing the non-Hermitian and non-reciprocal couplings, it is desirable to explore the connections between the other EPs, such as Weyl exceptional rings and higher-order EPs, and skin effect in PCs. In addition, it is interesting to explore the position-guided skin effect, where the fields of the skin mode can distribute at the given edge or corner.

## Note added

In revising the manuscript, we become aware of a new preprint reporting the observation of GDSE in the mechanical system[57], and another new preprint focusing on the observation of dynamical degeneracy splitting for the GDSE in the acoustic system[58].

## Methods

### Theory and simulation

In theory, we use the spectral function to show bulk Fermi arcs in Fig. 1b, which can be calculated by the formula $A(E) = -\frac{1}{\pi} \text{Im} G^r(E)$, where $G^r(E)$ is the retarded Green function of the model and $E$ is chosen the real part of the eigenvalue. For the finite-size sample under square-shaped geometry in Fig. 1d, the number of the outermost sites is $L_x = L_y = 40$, while that is $L_1 = L_2 = 30$ for the sample under diamond-shaped geometry in Fig. 1f. In simulation, all the simulations are performed by the commercial COMSOL Multiphysics solver package, where the air density and velocity of sound are chosen as $1.29 \, \text{kg/m}^3$ and $345 \, \text{m/s}$, respectively. We focus on the dipole mode of the cavity along the $z$ direction, so extract the pressure field from the cavity with the positions of $h/4$ or $3h/4$. In Fig. 3c, f, the 1D PC ribbon includes 31 cavities, and the defined boundary length contains 8 cavities nearest to lower ($L = 8$) and upper boundaries, respectively. The designed loss is induced by the imaginary part of sound velocity with value 50i m/s in coupling the corresponding waveguide (Supplementary S-II). In fact, the global loss is inevitable in experiment, so it is considered in Figs. 3d, g, S9b, S11a, S11c, S11e, and S11g in simulation. The global loss with 4.3i m/s on each cavity can induce some decay in all the states, but not affect the existence of skin effect.

### Sample and experiment

The experimental sample is fabricated by 3D printing technology, where the wall thicknesses of cavities and waveguides are set as 2 mm. To prevent structural deformation, we added additional support on the bottom of each cavity, which are shown as small solid-square rods in Fig. 2a. The designed loss on corresponding waveguide is realized by two holes sealed with the sound-absorbing sponges. Appropriate size of sponge can increase designed loss and reduce frequency shift. In order to obtain the uniform loss on each waveguide, the sponge in each hole should be consistent. The top of each cavity is provided with a lid for easy detection. To reduce the measurement error, the excited source (with diameter 6 mm) is embedded into the lid and the detector ($3 \times 1.8 \times 2 \, \text{mm}^3$) is small. Both the source and detector are connected to the network analyzer (E5061B 5Hz-500MHz). Response signals (forward transmission coefficient S$_{21}$) are measured at the concerned frequency range, where the scanning step of frequency $f$ is 2.25 Hz. All the pressure responses can be well detected, as discussed in Supplementary S-VIII with Fig. S16. In particular, the real parts of the frequencies for our PC sample under periodic boundary conditions are the same to those under open boundary conditions. So we can obtain the real parts of band dispersions by Fourier transforming the measured fields of practical PC samples. In Fig. 2c the bulk Fermi arcs, in Fig. 2e, f the bulk band dispersions, and in Fig. 3e, h the projected band dispersions are obtained by Fourier transforming the corresponding measured fields. In Fig. 3 (h), the source is placed in the center cavity of 5th row away from the lower (upper) boundary. In Fig. 4d of the response spectra, the source and detector are positioned at points 1 and 2 (3 and 4) marked in Fig. 4a for the skin modes and conventional

bulk modes, respectively. In Fig. 4e, f, the measured pressure field distributions for fixed frequencies are obtained by measuring the field of the cavity at $3h/4$, when placing the source at the top of the same cavity. Since the field response is different for different frequency, it is difficult to directly observe the spatial distribution of eigenstates $W(j)$ in experiment.

## Data availability

The data that support the plots within this paper and other findings of this study are available from the corresponding author upon reasonable request.

## Code availability

The codes that support the plots within this paper and other findings of this study are available from the corresponding author upon reasonable request.

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

## Acknowledgements

We thank Zhesen Yang for the helpful discussions. This work is supported by the National Key R&D Program of China (Nos. 2022YFA1404500, 2022YFA1404900, 2018YFA0305800), National Natural Science Foundation of China (Nos. 11890701, 11974120, 11974005, 11974262, 12074128, 12222405), and Guangdong Basic and Applied Basic Research Foundation (Nos. 2019B151502012, 2021B1515020086, 2022B1515020102).

## Author contributions

W.D. and Z.L. conceived the original idea. Q.Z., J.W., W.D., Z.P., and J.L. did the theoretical analysis and designed the structures. J.W., Q.Z., W.D., X.H., and M.K. performed the experiments. W.D., Q.Z., J.W., and Z.L. analyzed the data and wrote the manuscript. Z.L. and M.K. supervised the project. All authors participated in discussions and reviewed the paper.

## Competing interests

The authors declare no competing interests.
