## [Peer Review File · Nature Communications]

REVIEWER COMMENTS

Reviewer #2 (Remarks to the Author):

The authors present the first experimental demonstration of “geometry-dependent skin effect (GDSE)” and the underlying exceptional points (EPs) for acoustic waves in a non-Hermitian phononic crystal (PC). The experimental proofs are based on recent theoretical discoveries appeared in refs. 43-44. The experiments are performed on a 3D printed PC constituted by an array of cavities connected by lossy waveguides.

The EPs and corresponding GDSE are first discussed in the frame of a non-Hermitian tight-binding Hamiltonian on a square lattice. Then, the experiments evidence the Fermi arc joining the EPs and the corresponding GDSE in 1D ribbon under diamond-stripe geometry and finite sample under diamond-shape geometry. In contrast, the skin effect does not show up in 1D ribbon under square-stripe geometry and finite sample under square-shape geometry.

Although the theoretical principles were published in recent papers, the experimental results presented in this manuscript are very interesting and worth being published. Some issues could be further discussed or clarified.

- The spectra associated with the tight-binding Hamiltonian (Fig. 1) displays both positive and negative $\text{Im}(E)$ while the $\text{Im}(E)$ is positive in the spectra of the phononic crystal (Fig. 2). Please discuss this difference and in particular the possibility of $\text{Im}(E)$ negative in the absence of gain in the system.
- The spectra of the periodic structure is not easy to see from Fig. 1. It may be worth to adopt in SM other complementary presentations (for example a set of dispersion curves (real and imaginary parts) as a function of k_x for a series of values of k_y). Also, the discussion of Fig. 2(f) needs more explanation.
- Can the authors give a reference to a paper where this or similar tight-binding Hamiltonians have been proposed.
- Figure 3: A sketch of the 1D PC ribbon may be added. A short discussion can be given about the experimental procedure to measure the localization of the skin modes.
- More generally, what is the distance over which an excited field can be detected in Figs. 3 and 4?

Reviewer #3 (Remarks to the Author):

The non-Hermitian skin effect caused by the failure of the conventional bulk boundary correspondence promotes the modification of the Bloch band theory. Theoretical and experimental research on correspondence in non-Hermitian system is an important research direction at present. The author confirmed the exceptional points connected by the bulk Fermi arcs, and the 2D skin effects with the geometry dependence through acoustic experiment. This work is an important supplement to the current non-Hermitian skin effect theory and experimental verification which is quite nice.

I have several comments for the authors

1. Since the authors proposed geometry dependent skin effect, in addition to the current studied diamond and square lattice, is there any general rule to evaluate the dependence on the skin effect? This can be quite important and essential.
2. What's the sensitivity of EPs to temperature?
3. In page 4, the authors said "thus is the GDSE satisfying the volume law". It's necessary to explain the volume law here.
4. What's the origin of non-Hermitian Hamiltonian in (S3)?
5. For 2D phononic lattice with skin effect, is it possible to realize position-guided skin effect? Such as one can design the desired localization position at a given edge or corner. If not, is there any perspective on this problem?

Reviewer #4 (Remarks to the Author):

Motivated by a recent theoretical proposal, the authors examined the geometry dependent skin effect (GDSE) induced by exceptional points (EP) in 2D phononic systems. To observe GDSE, two different samples with square-shape and diamond-shape, respectively, are studied. It is found that only the sample with diamond shape exhibits skin effect. The authors additionally examined one-dimensional structures and further examined GDSE.

Experimental demonstration of GDSE induced by EP predicted recently is an interesting and timely topic. However, the description of the key theoretical idea and experiment is not clear. Also, I am not quite sure that the examination of just two different sample shapes is enough to claim the confirmation of GDSE induced by EPs.

Let me give a couple of additional questions/comments which might be useful to improve the paper.

(1) The relation between EP and GDSE should be clearly explained. Although the central idea was already proposed in a recent theory paper, as it is the key point of the present work, the related idea should be clarified before showing the experiments. What is the fundamental difference between the square shape and diamond shape in terms of GDSE? Is it enough to examine just these two shapes to demonstrate GDSE? If it is the case, what is the fundamental reason for that?

(2) Also, what is the second order EP? Does the GDSE depend on the order of EP?

(3) What is the meaning of the spectral area? It should be clearly defined. Also, does nonzero spectral area always give GDSE? Is EP the only source for nonzero spectral area? If nonzero spectral area has different origin, not related to EP, does it also give GDSE?

(4) In the context of phononic systems studied in this work, I am not quite sure how the authors confirmed i) the existence of EP and its topological charge including its magnitude and sign, ii) the presence of Fermi arc? The paper largely depends on the proposed theory and the tight-binding model in the manuscript. However, I think it is important how these properties can be confirmed in the current experimental setup. Also, is the observed skin effect purely originating from EP? Skin effect is generally expected in non-Hermitian systems. How can you show that the observed skin effect is completely induced by EP?

(5) In line 141, it is unclear what is the meaning of "a spectral loop winding the projection of EP."

(6) According to what is written in the introduction, EP and GDSE are properties of 2D non-Hermitian system. Why GDSE also appears in 1D ribbon structure? Is the observed GDSE in ribbon structure also related to EP? It may not be true. Also, it seems to be better to add one figure describing the 1D ribbon structure in figure 3.

(7) In lines 186-187, it is stated that "The eigenstates denoted by gray dots are the edge states induced by open boundaries," I do not see where the gray dots are.

(8) In line 198, T34 should be clearly defined.

Response to Reviewer #2:

The authors present the first experimental demonstration of “geometry-dependent skin effect (GDSE)” and the underlying exceptional points (EPs) for acoustic waves in a non-Hermitian phononic crystal (PC). The experimental proofs are based on recent theoretical discoveries appeared in refs. 43-44. The experiments are performed on a 3D printed PC constituted by an array of cavities connected by lossy waveguides.

The EPs and corresponding GDSE are first discussed in the frame of a non-Hermitian tight-binding Hamiltonian on a square lattice. Then, the experiments evidence the Fermi arc joining the EPs and the corresponding GDSE in 1D ribbon under diamond-stripe geometry and finite sample under diamond-shape geometry. In contrast, the skin effect does not show up in 1D ribbon under square-stripe geometry and finite sample under square-shape geometry.

Although the theoretical principles were published in recent papers, the experimental results presented in this manuscript are very interesting and worth being published. Some issues could be further discussed or clarified.

Reply: We thank Reviewer #2 for the careful reading and positive comments. We have addressed the following issues point by point, and the manuscript has been improved accordingly.

- The spectra associated with the tight-binding Hamiltonian (Fig. 1) displays both positive and negative $\text{Im}(E)$ while the $\text{Im}(E)$ is positive in the spectra of the phononic crystal (Fig. 2). Please discuss this difference and in particular the possibility of $\text{Im}(E)$ negative in the absence of gain in the system.

Reply: Thanks for your valuable comment. In the phononic crystal, the positive (negative) imaginary part of frequency indicates the attenuation (amplification) of acoustic field. When considering the loss, the intrinsic resonant frequency of each cavity will be influenced and appear global positive imaginary part which indicates attenuation. In the corresponding tight-binding model, such global loss can be described by adding positive imaginary part on the site energy, which will integrally lift the imaginary part of energy and make it positive. Since shifting the imaginary part of energy does not affect the topology property of the band, we do not consider this term in the tight-binding Hamiltonian for simplicity.

Since the negative imaginary part of frequency means the amplification of acoustic field, it is impossible to achieve that in the absence of gain in the phononic crystals.

The related discussion has been added in the main text of the revised manuscript.

- The spectra of the periodic structure is not easy to see from Fig. 1. It may be worth to adopt in SM other complementary presentations (for example a set of dispersion curves (real and imaginary parts) as a function of k_x for a series of values of k_y). Also, the discussion of Fig. 2(f) needs more explanation.

Reply: Following this helpful suggestion, we provide the detailed spectra of the tight-binding model under periodic boundary conditions in Fig. R1, including the dispersion curves (real and imaginary parts) along high symmetry lines, as a function of k_x for fixed k_y , and as a function of k_y for fixed k_x . We also provide the detailed spectra of the phononic crystal in Fig. R2. These results clearly show the exceptional points and bulk Fermi arcs existed in the tight-binding model and the phononic crystal.

Figures R1 and R2 with corresponding discussions have been added in Supplementary Material.

Fig. R1. Bulk band dispersions of the tight-binding model for different routes. **a**, Spectral area under periodic boundaries. **b**, Schematic of routes in momentum space. The black dashed line encloses the first Brillouin zone. Red (light blue) spheres denote the position of exceptional points with +1 (-1) topological charge. Yellow, cyan, blue and green lines represent the chosen routes 1-4, respectively. **c-f**, Dispersion curves containing both real

(solid lines) and imaginary parts (dashed lines) along the routes 1-4, respectively. Degeneracy of the bulk Fermi arc emerges in **c**, while degeneracies of exceptional points occur in **d** and **e**.

Fig. R2. Bulk band dispersions of the phononic crystal for different routes. **a-d**, Dispersion curves containing both real (solid lines) and imaginary parts (dashed lines) along the route 1-4 in momentum space shown in Fig. R1b. Degeneracy of the bulk Fermi arc emerges in **a**, while degeneracies of exceptional points occur in **b** and **c**.

- Can the authors give a reference to a paper where this or similar tight-binding Hamiltonians have been proposed.

Reply: To our best knowledge, up to now the non-Hermitian Hamiltonian in this manuscript has not been proposed, but its Hermitian part has been discussed in our previous paper [Phys. Rev. Appl. 12, 024007 (2019)]. We have cited it with related discussion in the revised manuscript.

- Figure 3: A sketch of the 1D PC ribbon may be added. A short discussion can be given about the experimental procedure to measure the localization of the skin modes.

Reply: Thanks for your helpful suggestion. In the revised manuscript, we add the sketch of the 1D PC ribbon shown in Fig. R3a. We also add the photographs of upper and lower boundaries for the experimental measurement of the skin modes, as shown in Fig. R3b. In experiment, we place a source (red star) near the upper boundary to excite the skin modes there. To avoid the edge states located at the sublattice that is the same to the outermost

cavity, we measure the response field of the second row of cavities near the upper boundary (magenta rectangle). The measured band dispersion of skin modes localizing at the upper boundary can be obtained by the Fourier transforming the measured field. The localization degree of the skin modes is revealed by the intensity of the Fourier transformation denoted by color. Similarly, by placing the source (blue star) and measuring the field (blue rectangle) near the lower boundary, the band dispersion of skin modes at the lower boundary is obtained.

In the revised manuscript, the above discussion has been given, and Figs. R3a and R3b have been added as Figs. 3a and 3b, respectively.

Fig. R3. The 1D PC ribbon and its experimental implementation. **a**, A sketch of the 1D PC ribbon. Insets show the experimental realization for upper and lower boundaries of the PC ribbon. **b**, Photographs of upper and lower boundaries for the experimental measurement of the skin modes. Magenta (blue) star represents the position of source at upper (lower) boundary, and the response fields are measured in the magenta (blue) rectangle.

-More generally, what is the distance over which an excited field can be detected in Figs. 3 and 4?

Reply: Thanks for your valuable question. As shown in Fig. R4a, the signal can be detected in all the cavities P_1 - P_3 , so the detectable distance of an excited field in Fig. 3 is 8 cavities at least. While in Fig. 4, the signal can be detected in the cavities P_4 and P_5 for all concerned frequencies, but in the cavity P_6 only for the higher frequencies, as shown in Fig. R4b, so the detectable distance is more than 8 cavities but less than 15 cavities.

The above discussion with Fig. R4 has been added in Supplementary Material.

Fig. R4. Observation of the pressure responses for different distances away from source. **a**, Photograph of lower boundary (upper panel) and the measured pressures at P_1 , P_2 and P_3 cavities (lower panel). **b**, Photograph of diamond-shaped sample (left panel) and the measured pressures at P_4 , P_5 and P_6 cavities (right panel).

Response to Reviewer #3:

The non-Hermitian skin effect caused by the failure of the conventional bulk boundary correspondence promotes the modification of the Bloch band theory. Theoretical and experimental research on correspondence in non-Hermitian system is an important research direction at present. The author confirmed the exceptional points connected by the bulk Fermi arcs, and the 2D skin effects with the geometry dependence through acoustic experiment. This work is an important supplement to the current non-Hermitian skin effect theory and experimental verification which is quite nice.

Reply: We thank Reviewer #3 for the careful reading and positive comments. We have addressed the following issues point by point, and the manuscript has been improved accordingly.

I have several comments for the authors:

1. Since the authors proposed geometry dependent skin effect, in addition to the current studied diamond and square lattice, is there any general rule to evaluate the dependence on the skin effect? This can be quite important and essential.

Reply: Thanks for your valuable question. Yes, there is a general rule to evaluate the geometry dependence on the skin effect. Whether the skin effect emerges at the open boundaries vertical to the n direction or not, depends on the winding number along the k_n direction, which is defined as

$$\nu(L, E_r) = \oint_L \frac{dk}{2\pi i} \cdot \nabla_k \ln \det[H(\mathbf{k}) - E_r],$$

where L is the closed straight line along the k_n direction in the first Brillouin zone, and E_r is the energy of reference. In addition, some spatial symmetries may guarantee the winding numbers along the related directions to be zero.

The spectral winding numbers of the square- and diamond- shaped samples are investigated in Supplementary Materials S-I. To further verify the general rule, we investigate the geometry-dependent skin effect in a parallelogram-shaped sample, in which the slope of the sloping open boundaries is 2, as shown in Fig. R5. The chosen route (straight lines in magenta) vertical to the direction of the sloping open boundaries is shown in Fig. R5a, which forms a closed loop in the first Brillouin zone. As shown in Fig. R5b, the spectral winding number is nonzero, leading to skin effect on the sloping open boundaries of the parallelogram-shaped sample (Fig. R5c). The winding number along the k_y direction is zero,

same to the case in the square-shaped sample, so no skin effect emerges at the horizontal open boundaries. Consequently, the geometry dependence on the skin effect can be generally evaluated by the spectral winding number along the corresponding direction.

The above discussions with Fig. R5 have been added in Supplementary Material.

Fig. R5. Skin effect under parallelogram-shaped geometry with sloping open boundaries possessing slope 2. **a**, Chosen route (magenta lines) vertical to the direction of the sloping open boundaries in the first Brillouin zone (dashed lines). The arrows denote the direction of the route. **b**, Complex spectra of chosen route in **a**. The winding number is +1 when the exceptional point (red sphere) is chosen as the energy of reference. **c**, Spatial distribution of eigenstates $W(j)$ under parallelogram-shaped geometry. The skin effect appears at the two sloping boundaries, and disappears at the two horizontal boundaries.

2. What's the sensitivity of EPs to temperature?

Reply: Thanks for your valuable question. The exceptional points (EPs) with topological charge ± 1 are stable in two dimensions [PRL 126, 086401 (2021)]. So the existence of EPs in our system is not sensitive to temperature, but their positions and frequencies may be changed by the temperature. Specifically, the temperature can affect the sound velocity, thereby acts on the band dispersion in the phononic crystal. In general, the relation between sound velocity c and temperature T ($^{\circ}\text{C}$) is estimated by $c = 331\sqrt{1 + T/273}$ m/s. When the temperature is confined at 15 $^{\circ}\text{C}$, 24 $^{\circ}\text{C}$ and 33 $^{\circ}\text{C}$, the sound velocity is about 340 m/s, 345 m/s and 350 m/s, respectively. Figure R6 gives the band dispersions near the EPs for these three temperatures. One can see that the EPs emerge in all these cases, and their frequencies lift as the temperature rises. As a result, the temperature mainly causes a frequency drift, but does not affect the existence of EPs.

The above discussions with Fig. R6 have been added in Supplementary Material.

Fig. R6. Band dispersions near the EPs for different temperatures. **a-c**, At the temperatures 15 °C, 24 °C and 33 °C, respectively. In our experiment, the temperature is confined at 24 °C.

3. In page 4, the authors said “thus is the GDSE satisfying the volume law”. It’s necessary to explain the volume law here.

Reply: Thanks for your helpful suggestion. We have added the explanation of the volume law in the revised manuscript, as: “thus is the GDSE satisfying the volume law, i.e., the number of skin modes increases in proportion to the increase in the volume of the system.”

4. What’s the origin of non-Hermitian Hamiltonian in (S3)?

Reply: Thanks for your helpful question. The non-Hermitian Hamiltonian in Eq. (S3) is the same to the one of Eq. (1) in the main text, which originates from the non-Hermitian square lattice model shown in Fig. 1a. To our best knowledge, up to now this non-Hermitian Hamiltonian has not been proposed, but its Hermitian part has been discussed in our previous paper [Phys. Rev. Appl. 12, 024007 (2019)]. We have cited it with related discussion in the revised manuscript.

5. For 2D phononic lattice with skin effect, is it possible to realize position-guided skin effect? Such as one can design the desired localization position at a given edge or corner. If not, is there any perspective on this problem?

Reply: Thanks for your constructive question. So far, the bulk states in the 2D non-Hermitian system can exhibit edge skin effect or corner skin effect, dependent on the nonzero spectral winding numbers along the related directions. However, it is an open question that how the fields of the skin mode distribute at the given edge or corner. In principle, the localization position of the skin mode would be related with the symmetry and microstructure of the given edge or corner. So the position-guided skin effect may be realized by modifying the

boundary case by case. We will keep exploring this interesting phenomenon in the future investigation.

We have added the related discussion in the last paragraph of the revised manuscript.

Response to Reviewer #4:

Motivated by a recent theoretical proposal, the authors examined the geometry dependent skin effect (GDSE) induced by exceptional points (EP) in 2D phononic systems. To observe GDSE, two different samples with square-shape and diamond-shape, respectively, are studied. It is found that only the sample with diamond shape exhibits skin effect. The authors additionally examined one-dimensional structures and further examined GDSE.

Experimental demonstration of GDSE induced by EP predicted recently is an interesting and timely topic. However, the description of the key theoretical idea and experiment is not clear. Also, I am not quite sure that the examination of just two different sample shapes is enough to claim the confirmation of GDSE induced by EPs.

Reply: We thank Reviewer #4 for the careful reading and critical comments. The key theoretical idea has been further described in detail in the following Replies (1), (3) and (6) with Figs. R8 and R9. And the experiment is further discussed in Reply (5). In particular, we have added the observation of the real parts of bulk band dispersion (Fig. R10), which together with the observation of the bulk Fermi arcs verifies the existence of EPs.

According to the definition of GDSE, the square- and diamond-shaped geometries do be enough to demonstrate GDSE, as discussed in Reply (1). We have further fabricated the finite-size PC sample under square-shaped geometry, confirming no skin effect in this geometry (Fig. R7), thus the cases with and without skin effect are both examined in experiment. At this time, we have corroborated the phononic system possessing EPs has the GDSE.

Figure R7 with related discussions has been added in Supplementary Material. We have addressed the following issues point by point, and the manuscript has been significantly improved accordingly.

Fig. R7. No skin effect in the finite-size PC sample under square-shaped geometry. **a**, Schematics of the finite-size PC sample. **b**, Spectral area (red dots) for the square-shaped PC. Blue shadow area denotes the spectrum under periodic boundaries. **c**, Real part of eigenfrequency spectrum. The color represents the degree of localization for each eigenstate on open boundaries. Inset: eigenpressure field of the edge mode. **d**, Measured response spectra at boundary (T_{12}) and bulk (T_{34}) normalized by their maximum values, which both show the bulk modes. **e**, **f**, Measured pressure field distributions at 4003 Hz and 4160 Hz, revealing the bulk and edge modes, respectively. No skin modes are observed in the square-shaped PC.

Let me give a couple of additional questions/comments which might be useful to improve the paper.

(1) The relation between EP and GDSE should be clearly explained. Although the central idea was already proposed in a recent theory paper, as it is the key point of the present work, the related idea should be clarified before showing the experiments. What is the fundamental difference between the square shape and diamond shape in terms of GDSE? Is it enough to examine just these two shapes to demonstrate GDSE? If it is the case, what is the fundamental reason for that?

Reply: Thanks for your helpful suggestions and questions. The relation between EP and GDSE is that the EP with nonzero topological charge can guarantee the emergence of the GDSE, but vice versa is not. A brief outline of the relation is illustrated in Fig. R8a. Firstly (Step I), the EPs host nonzero topological charges. Secondly (Step II), the nonzero topological charges lead to the nonzero spectral winding numbers along some straight lines

(directions) in the first Brillouin zone, which is equivalent to nonzero spectral area. Thirdly (Step III), the ribbons under open boundaries along the direction with nonzero spectral winding numbers host the skin effect, while those with zero winding number do not, giving rise to the GDSE in the 1D ribbons. Finally (Step IV), the GDSE in the 1D ribbons can directly result in the GDSE in the finite-size samples with fully open boundaries. It is noted that the EPs are not the must for the topological charges (Step I is single arrow), while the others are equivalent to each other (Step II-IV are double arrows).

Our lattice model is a concrete example to exhibit this relation. Figure R8b displays a pair of EPs connected by the bulk Fermi arc. The spectra loop winds the EP (red sphere) in an anticlockwise direction (arrow), indicating +1 topological charge of EP, as shown in Fig. R8c. Figure R8d demonstrates that the +1 charge of EP leads to nonzero spectral winding numbers along the k_2 direction. As shown in the left panel of Fig. R8d, when choosing the closed route $A \rightarrow A' \rightarrow B \rightarrow B' \rightarrow A$ encloses two EPs with +1 charge, the spectral winding number is the same to the sum of the topological charge of these two EPs. The winding number of this route is also equal to the sum of winding numbers for straight line loops $AA'BB'$ (magenta), $A'B$ (gray) and $B'A$ (cyan). Due to the mirror symmetry M_x satisfying $M_x H(k_x, k_y) M_x^{-1} = H(-k_x, k_y)$, the spectral winding number of route $A'B$ satisfies

$$\begin{aligned} \nu_{L,EP} &= \frac{1}{2\pi i} \oint_{L_{A'B}} dk_x \partial_{k_x} \ln \det[H(k_x, k_y)] \\ &= \frac{1}{2\pi i} \oint_{L_{A'B}} dk_x \partial_{-k_x} \ln \det[H(-k_x, k_y)] \\ &= -\nu_{L,EP} = 0, \end{aligned}$$

where $L_{A'B}$ indicates the integral along route $A'B$. Similarly, due to the mirror symmetry M_y , the spectral winding number of line $B'A$ is also zero. So the winding number of straight line $AA'BB'$ is equal to +2, attributing to the charges of these two EPs. Since the straight line $AA'BB'$ is the result of repeating line MN twice, the winding number of line MN is equal to +1, half of the winding number of $AA'BB'$. Correspondingly, the complex spectrum of line $AA'BB'$ (magenta) goes anticlockwise around the energy of EP twice, forming the spectral loop, while those of lines $A'B$ (gray) and $B'A$ (cyan) form spectral lines, as shown in the right panel of Fig. R8d. Therefore, the winding numbers along the k_x and k_y directions are zero, but those along the k_2 direction are nonzero. By a similar way, one can see that the nonzero winding numbers also exist along the k_1 direction. These results make the ribbons under open boundaries along the x and y directions have no skin effect, and the ribbons under open boundaries along the 1 and 2 directions exhibit the skin effect, giving rise to the GDSE in the

1D ribbons, as shown in Fig. R8e. Naturally, the skin effect emerges in the finite-size sample under the same open boundaries to the ones in the ribbons. So the skin effect disappears in the sample under square-shaped geometry, and appears under diamond-shaped geometry, thus is the GDSE, as shown in Fig. R8f. Consequently, the EP with nonzero topological charge can ensure the GDSE.

Fig. R8. Relation between EPs and GDSE. **a**, A brief outline of the relation. The double arrows denote the equivalency, while the single arrow means a sufficient but not necessary

condition. **b**, Bulk band dispersion near EPs. **c**, Left panel: A schematic of momentum space. The area enclosed by dash lines is the first Brillouin zone. Red (blue) spheres denote EPs with +1 (−1) topological charge. Right panel: Spectral loop on the complex plane plotted along the anticlockwise direction of the black circle route around the EP in the left panel. **d**, Left panel: Three chosen routes $AA'BB'$, $A'B$ and $B'A$ are denoted by magenta, gray and light blue lines, respectively. Right panel: Complex spectra of these three routes in the left panel. The blue background denotes the spectral area under periodic boundaries. **e**, Spatial distributions of $W(j)$ in the ribbons under square- (left panel) and diamond- (right panel) stripe geometries. **f**, Spatial distributions of $W(j)$ in the finite-size samples under square- (left panel) and diamond- (right panel) shaped geometries.

The fundamental difference between square-shaped and diamond-shaped geometries is that their open boundaries are in different directions, and more importantly, the spectral winding numbers along these directions are different, leading to GDSE, as shown in Fig. R8e. Specifically, the open boundaries of square-shaped geometry are in the x and y directions, and the spectral winding numbers of the straight lines along the k_x and k_y directions are zero, thus no skin effect appears in the sample under square-shaped geometry. However, the open boundaries of diamond-shaped geometry are in the 1 and 2 directions, and the spectral winding numbers along the k_1 and k_2 directions can be nonzero, guaranteed by the EPs, so the sample under diamond-shaped geometry exhibits the skin effect.

The GDSE is the phenomenon that there is at least one geometry with fully open boundary, in which the skin effect does not appear [Nat. Commun. 13, 2496 (2022)]. According to this definition, the square- and diamond-shaped geometries do be enough to demonstrate GDSE. We have further examined that no skin effect appears in the finite-size phononic crystal sample under square-shaped geometry, as shown in Fig. R7, thus the cases with and without skin effect are both confirmed in experiment. The other shape can also exhibit the GDSE, such as the sample under parallelogram-shaped geometry with sloping open boundaries possessing slope 2, as calculated in Fig. R9. No skin effect emerges at the horizontal open boundaries, same to the case under square-shaped geometry. But the skin effect appears at the sloping open boundaries, owing to the nonzero winding numbers along the corresponding direction.

The main discussions, including the relation between the EP and GDSE, the difference between square-shaped and diamond-shaped geometries, have been added in the main text.

And Figs. R8 and R9 with corresponding descriptions have been added in Supplementary Material.

Fig. R9. Spatial distribution of eigenstates $W(j)$ under parallelogram-shaped geometry with sloping open boundaries possessing slope 2. The skin effect appears at the two sloping boundaries, and disappears at the two horizontal boundaries.

(2) Also, what is the second order EP? Does the GDSE depend on the order of EP?

Reply: Thanks for your valuable questions. The EPs are singular points at which two or more eigenvalues, and their corresponding eigenvectors, coalesce and become degenerate [Science 363, 42 (2019)]. The second-order EP means that the degeneracy is two-fold. As discussed in the Reply (1), the nonzero topological charges of EPs can guarantee the emergence of the GDSE, which however has no direct connection with the order of EPs. In fact, there exist topologically trivial second-order EPs [PRL 126, 086401 (2021)], which cannot guarantee the GDSE in the system.

The related discussion has been added in the main text of the revised manuscript.

(3) What is the meaning of the spectral area? It should be clearly defined. Also, does nonzero spectral area always give GDSE? Is EP the only source for nonzero spectral area? If nonzero spectral area has different origin, not related to EP, does it also give GDSE?

Reply: Thanks for your helpful suggestions and questions. As discussed in Ref. [Nat. Commun. 13, 2496 (2022)], the spectral area is referred to the area covered by energy $E_m(\mathbf{k})$ on the complex plane, where m denotes the band index and \mathbf{k} is the wave vector in the first Brillouin zone. The nonzero spectral area is equivalent to the spectral winding numbers along some straight lines in the first Brillouin zone, thus always gives GDSE. The nonzero spectral area originates from the nonzero topological charge of generic point \mathbf{k}_r in the first Brillouin zone, which can be degenerate or not. The topological charge is calculated as

$$\nu_m(\mathbf{k}_r) = \oint_{\Gamma_{\mathbf{k}_r}} \frac{d\mathbf{k}}{2\pi i} \cdot \nabla_{\mathbf{k}} \ln \det[H(\mathbf{k}) - E_m(\mathbf{k}_r)],$$

where $E_m(\mathbf{k}_r)$ is the energy of the m -th band at the generic point \mathbf{k}_r . EP with nonzero topological charge is the typical generic point, but not the all, thus is not the only source for the nonzero spectral area.

The related discussion has been added in the main text of the revised manuscript.

(4) In the context of phononic systems studied in this work, I am not quite sure how the authors confirmed i) the existence of EP and its topological charge including its magnitude and sign, ii) the presence of Fermi arc? The paper largely depends on the proposed theory and the tight-binding model in the manuscript. However, I think it is important how these properties can be confirmed in the current experimental setup. Also, is the observed skin effect purely originating from EP? Skin effect is generally expected in non-Hermitian systems. How can you show that the observed skin effect is completely induced by EP?

Reply: Thanks for your valuable questions. Firstly, the EPs and bulk Fermi arcs are revealed by the real parts of the band dispersions, which are obtained by Fourier transforming the measured pressure fields. The bulk Fermi arcs have been shown in the Fig. 2c of the main text and Fig. S7 of the supplementary material. And the real parts of the band dispersions of EPs are further observed in Fig. R10. But the observation of the imaginary part of the band dispersion is difficult in the phononic crystals. Since the topological charge of EP is related with both the real and imaginary parts of the eigenvalues, the direct measurement of the topological charge of EP in the band dispersion still remain challenging. However, the observation of the bulk Fermi arcs can indirectly reflect that the connected EPs host nonzero and opposite topological charges, as discussed in Ref. [Science 359, 1009 (2018)]. In the revised manuscript, the above discussion has been added in the main text, and Fig. R10 has been added in Fig. 2.

Fig. R10. Bulk band dispersions of the phononic crystal for different routes. **a**, Schematic of routes in momentum space. The black dashed line encloses the first Brillouin zone. Red (light blue) spheres denote the position of exceptional points with +1 (−1) topological charge.

Cyan and blue lines represent the chosen routes 1 and 2, respectively. **b, c**, Simulated (white lines) and measured (color map) dispersion curves of routes 1 and 2, respectively.

Secondly, the observed skin effect is not purely originating from EPs. As discussed in Reply (1), the skin effect originates from the nonzero topological charges of the generic points, and EPs are just the typical ones but not the all of the generic points. So we cannot state that the observed skin effect is completely induced by EP. In fact, the existence of EP with nonzero topological charge can guarantee the emergence of the skin effect, but vice versa is not, as illustrated by the arrows in Fig. R8a. In experiments, we observe the GDSE and EP in the phononic system, verifying that the system with EP has the GDSE. To avoid the confusion of this connection between the EP and skin effect, we have revised the title as “**Observation of geometry-dependent skin effect in non-Hermitian phononic crystals with exceptional points**” and related discussions in the abstract and the main text.

(5) In line 141, it is unclear what is the meaning of “a spectral loop winding the projection of EP.

Reply: Thanks for your helpful question. The projection of EP means the position of EP on the complex plane. To avoid the confusion, we have revised it as “**a spectral loop winds the EP in an anticlockwise direction (arrow) on the complex plane.**”

(6) According to what is written in the introduction, EP and GDSE are properties of 2D non-Hermitian system. Why GDSE also appears in 1D ribbon structure? Is the observed GDSE in ribbon structure also related to EP? It may not be true. Also, it seems to be better to add one figure describing the 1D ribbon structure in figure 3.

Reply: Thanks for your helpful questions and suggestions. As discussed in Reply (1), because of the nonzero spectral winding numbers along some directions, the GDSE appears in 1D ribbon structure, and emerges in the finite-size sample under the same open boundaries to the ones in the ribbon. The EP with nonzero topological charge can guarantee the emergence of the GDSE in the 1D ribbon and the finite-size sample, but vice versa is not.

According to your suggestion, we have added one figure (as shown in Fig. R3) describing the 1D ribbon structure in Fig. 3 of the revised manuscript.

(7) In lines 186-187, it is stated that “The eigenstates denoted by gray dots are the edge states induced by open boundaries,” I do not see where the gray dots are.

Reply: We are sorry for the poor visibility of the gray dots. We have revised the range of $\text{Im}(f)$ in Fig. 4b to make the gray dots clearer, as shown in Fig. R9.

Fig. R11. Spectral area (red dots) for the diamond-shaped PC. Blue shadow area represents the spectrum under periodic boundaries. Gray dots denote the edge states induced by open boundaries.

(8) In line 198, T_{34} should be clearly defined.

Reply: Thanks for your helpful suggestion. In the revised manuscript, T_{34} has been defined as “ T_{34} is the response spectrum of bulk modes, where the source (detector) is located at point 3 (4) marked in Fig. 4a.”

REVIEWERS' COMMENTS

Reviewer #2 (Remarks to the Author):

The authors have carefully answered all the comments made by the referees. In the meantime, they have added several theoretical explanations and new illustrations. In my opinion, the paper can be accepted for publication in the present form.

There are a number of minor points mentioned below which the authors could take into consideration if they consider them useful.

In the manuscript:

- Line 90: give a reference to equation giving the discriminant number.
- New added section lines 96-104: the authors may revise this section both for the English presentation (i.e. "... but vice-versa is not", "... they are not the must ...") but more especially by noting that some terminologies introduced here are only defined later in the manuscript.
- line 120: here and at other places, the winding number is proposed to be calculated in straight lines in the Brillouin zone. It may be worth to clarify the meaning in relation to the closed loop appearing in the integrals defined in lines 90 or 123.

In SM:

- Figures S3 (d) and S3(e): for the sake of clarity, it would be helpful to add the points A, A', B, B' in the panel (e).
- Line 203: the meaning of reference energy (and perhaps its value in the following illustrations) can be clarified
- Figures S5 (a) and (b): the correspondence between the two panels could be made clearer.
- Line 355: "data" instead of "date"
- Section S-III and figure S10: the text explains what should be understood as routes 1, 2 and 3, but does not mention route 4 for panel (d). In addition, panels (c) and (d) have the same axes.

Reviewer #3 (Remarks to the Author):

The authors answered my comments well. in principle, I agree it for publication. just one minor suggestion, the authors may also mention skin effects in elastic wave systems in the introduction part which have important applications.

Reviewer #4 (Remarks to the Author):

I read the rebuttal letter and revised manuscript. First of all, I would like to thank the authors for their effort in preparing the detailed response letter which was very useful to clarify the confusing points and related concerns. In the response letter and revised manuscript, all my criticism and concerns are clearly answered and resolved. Thus, I recommend the publication of the revised manuscript.

Response to Reviewer #2:

The authors have carefully answered all the comments made by the referees. In the meantime, they have added several theoretical explanations and new illustrations. In my opinion, the paper can be accepted for publication in the present form.

There are a number of minor points mentioned below which the authors could take into consideration if they consider them useful.

Reply: We thank Reviewer #2 for the careful reading of our manuscript and recommending the publication. We have revised the manuscript according to these useful points.

In the manuscript:

- Line 90: give a reference to equation giving the discriminant number.

Reply: Thanks for your valuable suggestion. We have cited the Ref. [11] of the original manuscript in this place to give the definition of discriminant number.

- New added section lines 96-104: the authors may revise this section both for the English presentation (i.e. "... but vice-versa is not", "... they are not the must ...") but more especially by noting that some terminologies introduced here are only defined later in the manuscript.

Reply: According to this helpful suggestion, we have revised the related sentences as: "The relation between EP and GDSE is that the EP with nonzero topological charge can guarantee the emergence of the GDSE, but it is not vice versa. This relation is discussed in detail in Supplementary Material S-I D-G. The EPs host nonzero topological charges, but they are the sufficient and unnecessary condition for the topological charges. Then the topological charges ± 1 of EPs lead to the nonzero spectral winding numbers along some straight lines (directions) in the first BZ, which is defined in Supplementary Material S-I F. The nonzero winding number is equivalent to nonzero spectral area covered by energy $E_m(\mathbf{k})$ on the complex plane, where m denotes the band index and \mathbf{k} is the wave vector."

- line 120: here and at other places, the winding number is proposed to be calculated in straight lines in the Brillouin zone. It may be worth to clarify the meaning in relation to the closed loop appearing in the integrals defined in lines 90 or 123.

Reply: Thanks for your helpful suggestion. We have added the description of the closed loop when defining the winding number.

In SM:

- Figures S3(d) and S3(e): for the sake of clarity, it would be helpful to add the points A, A', B, B' in the panel (e).

Reply: Following this valuable suggestion, we have added the points A, A', B, B'.

- Line 203: the meaning of reference energy (and perhaps its value in the following illustrations) can be clarified

Reply: Thanks for your valuable suggestion. The reference energy means a reference point on the complex plane of energy. We have marked the reference energy in Fig. S4.

- Figures S5 (a) and (b): the correspondence between the two panels could be made clearer.

Reply: Thanks for your helpful suggestion. We have added some labelled points in these two panels to make the correspondence clearer.

- Line 355: “data” instead of “date”

Reply: Thanks for your careful reading. We have corrected the incorrect word.

- Section S-III and figure S10: the text explains what should be understood as routes 1, 2 and 3, but does not mention route 4 for panel (d). In addition, panels (c) and (d) have the same axes.

Reply: Thanks for your valuable suggestion. We have added “No degeneracy occurs in **d**.” in the text. Panels (c) and (d) have the same k_y axis but with different k_x , as shown in Fig. S2b.

Response to Reviewer #3:

The authors answered my comments well. in principle, I agree it for publication. just one minor suggestion, the authors may also mention skin effects in elastic wave systems in the introduction part which have important applications.

Reply: We thank Reviewer #3 for recommending the publication of our manuscript. Following this helpful suggestion, we have cited the references [Phys. Rev. Lett. 125, 118001 (2020); J. Phys. D: Appl. Phys. 54, 285302 (2021); Appl. Phys. Lett. 122, 222203 (2023)] for the skin effect in elastic wave systems in the revised manuscript.

Response to Reviewer #4:

I read the rebuttal letter and revised manuscript. First of all, I would like to thank the authors for their effort in preparing the detailed response letter which was very useful to clarify the confusing points and related concerns. In the response letter and revised manuscript, all my criticism and concerns are clearly answered and resolved. Thus, I recommend the publication of the revised manuscript.

Reply: We thank Reviewer #4 for recommending the publication of our manuscript.